# High Glucose Activates Prolyl Hydroxylases and Disrupts HIF-α Signaling via the P53/TIGAR Pathway in Cardiomyocyte

**DOI:** 10.3390/cells12071060

**Published:** 2023-03-31

**Authors:** Jian-Xiong Chen, Lanfang Li, Aubrey C. Cantrell, Quinesha A. Williams, Heng Zeng

**Affiliations:** Department of Pharmacology and Toxicology, School of Medicine, University of Mississippi Medical Center, Jackson, MS 39216, USA; jchen3@umc.edu (J.-X.C.);

**Keywords:** high glucose, prolyl hydroxylases (PHDs), HIF-1α, p53, TIGAR, glycolysis

## Abstract

The induction of hypoxia tolerance has emerged as a novel therapeutic strategy for the treatment of ischemic diseases. The disruption of hypoxic signaling by hyperglycemia has been shown to contribute to diabetic cardiomyopathy. In this study, we explored the potential molecular mechanisms by which high glucose (HG) impairs hypoxia-inducible factor-α (HIF-α) signaling in cardiomyocytes. The exposure of H9c2 cell lines to HG resulted in time- and concentration-dependent decreases in HIF-1α and HIF-2α expression together with an increase in prolyl hydroxylase-1,2 (PHD1 and PHD2) expression, the main regulators of HIF-α destabilization in the heart. The exposure of H9c2 cells to normal glucose (5.5 mM) and high glucose (15, 30, and 45 mM) led to dose-dependent increases in p53 and TIGAR and a decrease in SIRT3 expression. The pretreatment of H9c2 with p53 siRNA to knockdown p53 attenuated PHD1 and PHD2 expression, thus significantly enhancing HIF-1α and HIF-2α expression in H9c2 cells under HG conditions. Interestingly, pretreatment with p53 siRNA altered H9c2 cell metabolism by reducing oxygen consumption rate and increasing glycolysis. Similarly, pretreatment with TIGAR siRNA blunted HG-induced PHD1 and PHD2 expression. This was accompanied by an increase in HIF-1α and HIF-2α expression with a reduction in oxygen consumption rate in H9c2 cells. Furthermore, pretreatment with adenovirus-SIRT3 (Ad-SIRT3) significantly reduced the HG-induced expression of p53 and PHDs and increased HIF-1α levels in H9c2 cells. Ad-SIRT3 treatment also regulated PHDs-HIF-1α levels in the hearts of diabetic db/db mice. Our study revealed a novel role of the HG-induced disruption of PHDs-HIF-α signaling via upregulating p53 and TIGAR expression. Therefore, the p53/TIGAR signaling pathway may be a novel target for diabetic cardiomyopathy.

## 1. Introduction

Hyperglycemia is considered to be one of the major factors contributing to diabetic cardiomyopathy [1,2,3]. Hyperglycemia has been shown to exacerbate acute myocardial infarction (AMI), and the incidence of heart failure after AMI increases at an alarming rate in diabetic patients [4,5,6,7,8,9,10]. High mortality rates in diabetic patients with AMI might be caused by impaired ischemic pre-conditioning [11,12], larger infarct size, and increased incidence of the no-reflow phenomenon [13]. Therefore, it is urgent to identify potential therapeutic targets which can reduce the high mortality rate of diabetic patients with AMI. The induction of hypoxic tolerance to ischemia by reprogramming glycolytic metabolism via the prolyl hydroxylase-1 (PHD1)-hypoxia-inducible factor-2α (HIF-2α) pathway has emerged as a novel therapeutic strategy for the treatment of ischemic diseases [14,15,16,17]. There is increasing evidence that the diabetic heart has a reduced tolerance to ischemia [18]. Defective myocardial HIF-1α expression and impaired HIF-1α signaling have been demonstrated in diabetic patients and contribute to diabetic cardiomyopathy [19,20]. Furthermore, the hyperglycemia-induced degradation of HIF-1α contributes to the impaired response of cardiomyocytes to hypoxia [21]. We have shown that diabetes disrupts prolyl hydroxylase (PHD) and hypoxia-inducible factor-α (HIF-α) signaling, thus leading to the impairment of angiogenesis [22,23,24]. However, there is still much to uncover about the precise signaling mechanisms responsible for the impairment of hypoxic signaling, both in the diabetic heart and under hyperglycemic conditions.

Emerging evidence reveals that the level of the tumor suppressor protein p53 is elevated in human heart failure [25,26]. Furthermore, the accumulation of p53 demonstrated in heart failure has been attributed to cardiomyocyte senescence and myocardial microvascular rarefaction [27,28]. A study also shows that the upregulation of p53 suppresses capillary formation by reducing HIF-1α in the failing heart [29]. In contrast, knockout of p53 in mice attenuates doxorubicin-induced heart failure [30,31]. The specific knockout of p53 in endothelial cells (ECs) has also been shown to increase capillary density in pressure overload-induced heart failure [32]. TP53-induced glycolysis and apoptosis regulator (TIGAR) is a novel downstream target gene of p53 and a regulator of glycolysis and apoptosis whose expression level is regulated by p53. TIGAR is involved in various biological processes, including metabolism, apoptosis, the cell cycle, and cell death [33,34]. TIGAR is upregulated in cardiomyocytes upon exposure to hypoxia, whereas the knockout of TIGAR attenuates ischemic heart failure [35,36]. Our previous study reveals a novel role of p53/TIGAR in the regulation of glycolysis and angiogenesis in cardiomyocytes under high glucose conditions. Our data revealed that levels of p53 and TIGAR were upregulated in H9c2 cells exposed to high glucose and in the hearts of diabetic db/db mice [37]. So far, the direct link between p53/TIGAR and HIF-α signaling in diabetic cardiomyopathy or hyperglycemia has not been investigated.

In this study, we hypothesized that high glucose disrupts PHDs-HIF-α signaling by a mechanism involving the upregulation of p53/TIGAR which contributes to the reprogramming of cellular metabolism by altering oxygen consumption rate (OCR) and glycolysis. The exposure of H9c2 cells to high glucose (HG) enhanced PHDs and reduced HIF-α expression. The pretreatment of H9c2 cells with p53 siRNA or TIGAR siRNA to knockdown p53 or TIGAR blunted HG-induced PHD1 and PHD2 expression and reversed the impairment of HIF-α signaling. We conclude that the upregulation of p53/TIGAR promotes diabetic cardiomyopathy by a mechanism involving the disruption of the PHDs-HIF-α signaling pathway.

## 2. Methods and Materials

The authors declare that all supporting data are available within the article.

All protocols were approved by the Institutional Animal Care and Use Committee (IACUC) of the University of Mississippi Medical Center (Protocol ID: 1280C) and were consistent with the National Institutes of Health Guide for the Care and Use of Laboratory Animals (NIH Pub. No. 85–23, Revised 1996).

### 2.1. Experimental Mice

Male db+/− mice and diabetic db/db mice were purchased from the Jackson Laboratory (Bar Harbor, ME). We choose male diabetic mice because they had significant cardiac dysfunction at this age compared to female mice which had mild cardiac dysfunction. The experimental mice at the age of 10–12 months were divided into the following: (1) db+/− mice; n = 5; (2) db/db mice; n = 5; and (3) db/db+Ad-SIRT3 mice; n = 5. Experimental mice received an intravenous jugular vein injection of Ad-SIRT3 [Human Adenovirus Type5 (dE1/E3), CMV promoter] [1 × 10^9^ plaque-forming units (PFU)] (Vector Biolabs, Malvern, PA, USA) as we previously reported [37]. Immunohistochemical analysis revealed that SIRT3 was colocalized with troponin on cardiomyocytes and isolectin B4 on ECs in the adenovirus-SIRT3–treated db/db mouse hearts [37].

### 2.2. Cell Culture

Rat cardiomyocyte H9c2 (ATCC CRL1446) cell lines were grown in Dulbecco’s Modified Eagle Medium-low glucose (DMEM L-glucose, 5.5 mM from Sigma-Aldrich, St. Louis, MO, USA) with the addition of 10% fetal bovine serum (FBS, from Invitrogen, OR, USA), 2 mmol/L glutamine, 10^4^ × diluted 10,000 U/mL penicillin, and 10 mg/mL streptomycin (from Sigma-Aldrich).

### 2.3. Western Blot Analysis

Cultured cells or left ventricular samples were homogenized with an ice-cold RIPA buffer. Protein concentration was measured with the Bradford reagent (Sigma, B6916). The PVDF membranes were probed with antibodies specific to Sirtuin 3 (SIRT3) (Cell Signaling Technology, Danvers, MA, USA), p53 (Abcam, Waltham, MA, USA), TP53-induced glycolysis and apoptosis regulator (TIGAR) (Santa Cruz Biotechnology, Dallas, TX, USA), hypoxia-inducible factor-1α (HIF-1α), hypoxia-inducible factor-2α (HIF-2α) (Novus Bio, Littleton, CO, USA), prolyl hydroxylase domain protein-1 (PHD1) (Novus Bio, Littleton, CO, USA), prolyl hydroxylase domain protein-2 (PHD2) (Novus Bio, Littleton, CO, USA), transforming growth factor beta (TGF-β) (Santa Cruz Biotechnology, Dallas, TX, USA), vascular endothelial growth factor (VEGF), angiopoietin-1 (Ang-1) (Santa Cruz Biotechnology, Dallas, TX, USA), β-actin (Cell Signaling Technology, Danvers, MA, USA), or GAPDH (Cell Signaling Technology, Danvers, MA, USA). The membranes were then washed and incubated with an anti-rabbit or anti-mouse secondary antibody conjugated with horseradish peroxidase. Densitometries were analyzed using Image analysis software 6.0 (Bio-Red, Hercules, CA, USA).

### 2.4. RNA Interference

H9c2 cells were transfected with 25 nM TIGAR siRNA or p53 siRNA as well as control scramble siRNA (TriFecTaDsi DNA duplex, Integrated DNA Technologies, INC, Coralville, IA, USA) by using the TransIT-X2 Dynamic Delivery System (Integrated DNA Technologies) following the manufacturer’s instruction. Knockdown of p53 and TIGAR was confirmed by Western blot as we previously reported [37]. These gene knockdown cells were then exposed to high glucose (30 mM) for 72 h.

### 2.5. Adenoviral Vectors Transfection in H9c2 Cell Lines

The human SIRT3 adenoviral vector (Ad-SIRT3) or control GFP adenoviral vector (Ad-GFP) was infected into H9c2 cell lines. Briefly, H9c2 cells were infected by incubation with Ad-SIRT3 or Ad-GFP at two dosages of 0.5 × 10^6^ PFU/mL (1:5) or 1 × 10^6^ PFU/mL (1:10) for 48 h. Western blot analysis confirmed that Ad-SIRT3 incubation resulted in a dose-dependent increase in SIRT3 expression in H9c2 cells [37]. These infected cells were then exposed to high glucose (30 mM) for 72 h.

### 2.6. Metabolic Assays

Glycolysis was determined using the XF^e^24 extracellular flux analyzer from Seahorse Bioscience, following the manufacturer’s instructions. Briefly, cells were seeded in a TC-treated 24-well plate (V7-PS) at the density of 25,000 cells per well as determined by a pilot cell density assay. The next day, the cells were incubated in an unbuffered assay medium supplemented with various substrates as described below, at 37 °C in a non-CO_2_ incubator for 1 h prior to analysis. The unbuffered assay medium was supplemented with glutamine (2 mM) only. The baseline extracellular acidification rate (ECAR) was measured, followed by the sequential injection of the following compounds with indicated final concentrations: glucose (10 mM), oligomycin (1 μM), and 2-deoxyglucose (2-DG, 100 mM). The levels of basal glycolysis, glycolytic reserve, and glycolytic capacity were calculated from the raw data using the Seahorse report generator.

For the cellular mitochondrial stress test, the unbuffered assay medium was supplemented with glucose (10 mM), pyruvate (1 mM), and glutamine (2 mM). The baseline oxygen consumption rate (OCR) was measured, followed by a subsequent injection of the following inhibitors with indicated final concentrations: oligomycin (1 μM), cyanide p-trifluoromethoxy-phenylhydrazone (FCCP, 1 μM), and rotenone/antimycin A (0.5 μM). Basal respiration, maximal respiration, spare respiration, non-mitochondrial respiration, and proton leak were calculated from the raw data using the Seahorse report generator.

### 2.7. Immunofluorescence Analysis

H9c2 cells were fixed with 10% formalin for 15 min at room temperature (RT). After this, the cells were permeabilized with 0.2% Triton X-100 in PBS for 10 min. Non-specific binding was blocked via incubation with 10% FCS in PBS for 30 min at 37 °C in a humidified chamber. H9c2 cells were immunostained with fibroblast-specific protein-1 (FSP-1) primary antibodies (1:200) followed by incubation with secondary antibodies conjugated with fluorescein isothiocyanate (FITC) or Cy3 (1:500). The area percentage of fluorescence intensity was quantified at six random microscopic fields using Image J analysis software (Image J, NIH, Bethesda, MD, USA).

Left ventricular samples were embedded and frozen in OCT compound (4583; Sakura Finetek, Torrance, CA, USA), and 10 µm frozen sections were prepared. Sections were immunostained with Troponin and FSP-1 primary antibodies (1:200) followed by incubation with secondary antibodies conjugated with fluorescein isothiocyanate (FITC) or Cy3 (1:500). Photomicrographs were obtained with an Olympus BX51 microscope, a Q-Color5 digital camera and a Q-Capture Suite acquisition software (Olympus, Tokyo, Japan).

### 2.8. Statistical Analysis

Data are presented as mean ± SD. The assumption of normality in both comparison groups was determined by a normality test. Statistical significance was determined using one-way ANOVA followed by Tukey’s post hoc test for multiple comparisons using GraphPad Prism 8.1.1 software (GraphPad Software, La Jolla, CA, USA). *p* < 0.05 was considered statistically significant.

## 3. Results

### 3.1. High Glucose (HG) Upregulated PHDs and Reduced HIF-α Expression in H9c2 Cells

To explore the effects of HG on hypoxic signaling, the time- and dose-dependent expression of PHDs-HIF-α under normal-glucose (5.5 mM) or high-glucose conditions in H9c2 cell lines were examined. The exposure of H9c2 cell lines to HG (30 mM) for various time periods up to 72 h led to a gradual increase in PHD1 and PHD2 expression as compared to control normal glucose (5.5 mM). This was accompanied by the downregulation of HIF-1α expression (Figure 1). Furthermore, the exposure of H9c2 cell lines to various concentrations of HG (15, 30, and 45 mM) for 72 h resulted in a dose-dependent increase in PHD1 and PHD2 expression and the downregulation of HIF-1α and HIF-2α expression as compared to normal glucose (5.5 mM). The exposure of H9c2 cells to HG also caused a dose-dependent reduction in Ang-1 expression and upregulation of TGF-β expression as compared to normal glucose (5.5 mM) (Figure 2A–C).

### 3.2. Pretreatment with p53 siRNA Reduced Expression of PHDs and Improved HIF-α Signaling in H9c2 Cells under HG Conditions

To test the involvement of p53 in the HG-induced impairment of HIF-α signaling, the levels of p53 in H9c2 cells after exposure to different concentrations of glucose were first examined. As shown in Figure 3A, the exposure of H9c2 cell lines to various concentrations of HG (15, 30, and 45 mM) for 72 h led to a dose-dependent upregulation of p53 expression as compared to normal glucose (5.5 mM). The exposure of H9c2 cell lines to Mannitol (30 mM) had no effects on the levels of p53 expression [37].

Knockdown of p53 by p53 siRNA was confirmed by Western blot in our previous study [37]. As shown in Figure 3B, pretreatment of H9c2 with p53 siRNA significantly reduced HG-induced PHD1 and PHD2 expression in H9c2 cells. These were followed by a significant upregulation of HIF-1α and HIF-2α expression (Figure 3C). Pretreatment with p53 siRNA also significantly increased angiogenic growth factor Ang-1 and VEGF levels, whereas HG-induced TGF-β expression was blunted in H9c2 cells (Figure 3D).

### 3.3. Pretreatment with p53 siRNA Altered H9c2 Cell Metabolism under HG Conditions

We next examined whether the activation of p53 altered cell metabolism in H9c2 cells under HG conditions by measuring oxygen consumption rate (OCR) and glycolysis (ECAR) using the Seahorse analyzer. The exposure of H9c2 cells to HG for 72 h resulted in a significant increase in OCR, non-mitochondrial respiration, and proton leak (Figure 3E). Pretreatment with p53 siRNA significantly attenuated HG-induced OCR, non-mitochondrial respiration, and proton leak in H9c2 cells (Figure 3E). Furthermore, the exposure of H9c2 cells to HG led to an impairment of glycolysis and glycolytic capacity. Pretreatment with p53 siRNA significantly improved the HG-induced impairment of glycolysis and glycolytic capacity in H9c2 cells (Figure 3F).

### 3.4. Pretreatment with TIGAR siRNA Reduced PHDs and Enhanced HIF-α Signaling in H9c2 Cells

To further elucidate the potential signaling pathway of p53 in the HG-induced impairment of HIF-α signaling, the levels of TIGAR, downstream of p53 signaling, were examined in H9c2 cells under different glucose concentrations. As shown in Figure 4A, the exposure of H9c2 cell lines to various concentrations of HG (15, 30, and 45 mM) for 72 h resulted in a dose-dependent upregulation of TIGAR expression as compared to normal glucose (5.5 mM). The exposure of H9c2 cell lines to mannitol (30 mM) had no effects on the levels of TIGAR expression [37].

Knockdown of TIGAR by TIGAR siRNA was confirmed in our previous study [37]. As shown in Figure 4B, pretreatment with TIGAR siRNA significantly reduced HG-induced PHD1 and PHD2 expression, accompanied by significant increases in HIF-1α and HIF-2α expression in H9c2 cells (Figure 4C). Pretreatment with TIGAR siRNA also significantly blunted HG-induced FSP-1 expression (Figure 4D). Furthermore, pretreatment with TIGAR siRNA altered cell metabolism as evidenced by a significant reduction in HG-induced OCR, non-mitochondrial respiration, and proton leak in H9c2 cells (Figure 4E).

### 3.5. SIRT3 Regulates PHDs-HIF-α Signaling Pathway in H9c2 Cells and in Diabetic Hearts

Our previous study demonstrated that the overexpression of SIRT3 by Ad-SIRT3 treatment reduced p53 expression in diabetic hearts [37]. To further test the potential role of SIRT3 in HG-induced impairment of HIF-α signaling, we first examined the levels of SIRT3 expression in H9c2 cells under different glucose concentrations. As shown in Figure 5A, the exposure of H9c2 cell lines to various concentrations of HG (15, 30, and 45 mM) for 72 h led to a dose-dependent reduction in SIRT3 expression as compared to normal glucose (5.5 mM). The exposure of H9c2 cell lines to Mannitol (30 mM) had no effects on the levels of SIRT3 expression [37].

Next, we examined whether SIRT3 altered p53 and PHDs-HIF-α expression in H9c2 cells. H9c2 cell lines were infected with adenovirus-SIRT3 at different concentrations (1:5 or 1:10) or control Ad-GFP. The efficacy of Ad-SIRT3 transfection in H9c2 cell lines was demonstrated in our previous study [37]. As shown in Figure 5A, Ad-SIRT3 treatment resulted in a significant reduction in p53 expression in H9c2 cell lines. Furthermore, Ad-SIRT3 treatment significantly reduced PHD1 and PHD2 expression (Figure 5B), followed by a significant increase in HIF-1α expression (Figure 5C). Ad-SIRT3 treatment also significantly attenuated HG-induced FSP-1 expression in H9c2 cell lines (Figure 5D).

The regulatory role of SIRT3 in cardiac PHDs-HIF-1α signaling was further validated in the diabetic db/db mice. Treatment with Ad-SIRT3 significantly reduced levels of myocardial PHD1 and PHD2 expression, together with increased HIF-1α levels in the hearts of db/db mice (Figure 6A,B). Furthermore, treatment with Ad-SIRT3 significantly reduced myocardial FSP-1 expression in the hearts of db/db mice (Figure 6C).

## 4. Discussion

In this study, we demonstrated that high glucose disrupted the PHDs-HIF-α signaling pathway and altered cell metabolism via a mechanism involving the upregulation of p53 and TIGAR and the downregulation of SIRT3 expression. In vitro, pretreatment with p53/TIGAR siRNA or Ad-SIRT3 in cardiomyocytes blunted HG-induced the impairment of HIF-α signaling and reprogrammed cellular metabolism by reducing the oxygen consumption rate. In vivo, Ad-SIRT3 treatment resulted in the suppression of PHDs, followed by an increase in HIF-1α in diabetic db/db mice. Our data demonstrated for the first time that hyperglycemia and diabetes modulate hypoxic signaling and oxygen consumption rate by mechanisms involving disruption of PHDs-HIF-α via the upregulation of the p53/TIGAR signaling pathways.

HIF-α is a transcriptional activator that is expressed in response to hypoxia/ischemia [38,39]. HIF-1α and HIF-2α are two isoforms of HIF-α with high-sequence homology [40,41,42,43,44]. Germline deletion of either HIF-α subunit demonstrates that important and unique roles exist for both HIF-1α and HIF-2α. Specifically, HIF-1α is a pathway regulator of genes encoding cell metabolism [41,42,43,45], whereas HIF-2α functions to maintain mitochondrial homeostasis [46,47]. Hyperglycemia has been shown to reduce HIF-1α expression in human microvascular endothelial cells and rat proximal tubule cells [48,49]. Defective hypoxic signaling and the impairment of HIF-α expression have contributed to the dysfunction of pancreatic β cells in diabetic patients [50]. Reduced HIF-1α expression is also found in heart specimens of diabetic patients with unstable angina [20]. During myocardial ischemia/reperfusion, hyperglycemia reduced HIF-1α expression which was associated with enlarged infarction size. Interestingly, these changes are reversed by a return to normal glucose levels [19]. Our previous studies have demonstrated that defective hypoxic signaling and the loss of HIF-1α expression contribute to impaired angiogenesis and cardiac dysfunction in diabetes [22,23,24]. However, the underlying mechanisms by which diabetes and hyperglycemia impair hypoxic signaling in cardiomyocytes remains unknown. HIF-α stabilization is regulated by HIF-1α-prolyl-4-hydroxylases (PHDs), which target HIF-α for ubiquitination and proteasomal degradation [51,52,53]. So far, three isoforms of PHD (PHD1–3) have been identified. Among them, PHD2 is a major negative regulator for HIF-1α expression under hypoxia, whereas PHD1 is more active on HIF-2α than on HIF-1α [51,54,55]. In our present study, we found that the levels of PHD1 and PHD2 were significantly elevated in the hearts of diabetic db/db mice and in H9c2 cells exposed to high glucose conditions. This was accompanied by a significant reduction in both HIF-1α and HIF-2α levels, as well as down-streaming signaling, angiogenic growth factor Ang-1, and VEGF expression. Our data suggested that diabetes and hyperglycemia may disrupt hypoxic signaling via the activation of PHDs, which destabilize HIF-α proteins in diabetic hearts.

Currently, our knowledge of the involvement of PHDs in diabetes and hyperglycemic stress is still limited, and it will be important to examine the role of PHD isoforms under high-glucose conditions. Recent studies implicate PHD1 as the key regulator of oxygen conformance and as a modulator of hypoxic tolerance under ischemic conditions. The inhibition of PHD1 reduces oxygen consumption and mitochondrial oxidative stress and protects against muscle ischemic necrosis in a HIF-2α-dependent fashion [15]. This endogenous protection is caused, at least in part, by reprogramming the basal metabolism, in particular by a reduction in oxidative glucose metabolism, which is correlated with lower pyruvate dehydrogenase complex activity and restricts the entry of glycolytic intermediates into the tricarboxylic acid (TCA) cycle [15]. These findings implicate a novel role of PHD1 in the protection of ischemic muscle against oxidative damage and the induction of hypoxic tolerance through a mechanism involving the activation of a hibernating state and preserving mitochondrial integrity. Although diabetic hearts exhibit reduced tolerance to hypoxia, the underlying mechanism of this impairment is unknown. Our data demonstrate that PHD1 expression is significantly increased in diabetic db/db mouse hearts and in H9c2 cells exposed to high-glucose conditions and that this is accompanied by a dramatic decrease in the expression of HIF-2α. Intriguingly, H9c2 cells exposed to high glucose reprogrammed cellular metabolism by increasing the oxygen consumption rate (OCR) and proton leak while reducing glycolysis. These data implicate that the upregulation of PHD1, may lead to reducing cardiomyocyte tolerance to hypoxia and contribute to diabetic cardiomyopathy. Although our present study implicates a potential involvement of PHD1 in diabetic cardiomyopathy, the direct role of PHD1 in cardiomyocyte metabolic reprogramming of diabetic cardiomyopathy is lacking and warrants further investigation.

Our previous study also demonstrated that the expression of PHD1 and PHD2 was upregulated, whereas HIF-2α was downregulated in SIRT3KO mouse hearts [56]. Moreover, the inhibition of PHDs upregulated hypoxic signaling, improved glycolysis, and reduced oxygen consumption rate in endothelial cells. This led to a significant improvement in angiogenesis and cardiac function, suggesting that SIRT3 deficiency may impair hypoxic signaling [56]. However, whether an HG-induced reduction in SIRT3 contributes to the impairment of hypoxic signaling in diabetes remains unexplored. To test the direct link between SIRT3 and the disruption of PHDs-HIF-α signaling, H9c2 cells were transfected with adenovirus-SIRT3 and subsequently exposed to high-glucose conditions. Under HG conditions, SIRT3 levels were reduced in a dose-dependent manner. The overexpression of SIRT3 significantly attenuated the HG-induced upregulation of p53 and PHDs, followed by a significant increase in HIF-1α expression, suggesting a novel regulatory role of SIRT3 in hypoxic signaling under hyperglycemic conditions. Using diabetic db/db mice treated with Ad-SIRT3, we further validated that the overexpression of SIRT3 reduced the expression of myocardial PHDs, which were accompanied by increased HIF-1α levels in diabetic hearts. These in vivo results further support the regulatory role of SIRT3 in hypoxic signaling of the diabetic heart. Our present data also indicate that the p53/TIGAR axis appears to be the key signaling pathway modulating the PHDs-HIF-α pathway under hyperglycemic conditions since the knockdown of p53 and TIGAR resulted in a reduction in PHD expression and the upregulation of HIF-α expression.

In the present study, the exposure of H9c2 cells to HG altered cell metabolism as evidenced by significant reductions in glycolysis, glycolysis capacity, and glycolytic reserve, whereas the oxygen consumption rate, non-mitochondrial respiration, and mitochondrial proton leak were significantly increased. Importantly, the knockdown of p53 significantly increased glycolysis under high-glucose conditions and significantly suppressed HG-induced increases in oxygen consumption rate, non-mitochondrial respiration, and mitochondrial proton leak in H9c2 cells. Similarly, the knockdown of TIGAR reduced HG-induced increases in oxygen consumption rate, non-mitochondrial respiration, and mitochondrial proton leak. These data suggest that targeting the p53/TIGAR axis may alleviate hyperglycemia-induced cardiomyopathy via a mechanism involving the hypoxic signaling pathway and reprogrammed cell metabolism (Figure 6D).

## 5. Conclusions

In conclusion, our present study suggests that both diabetes and hyperglycemia disrupt the PHDs-HIF-α signaling pathway and reprogram cellular metabolism by a mechanism involving the upregulation of p53/TIGAR and the downregulation of SIRT3. The results from the present studies provide a foundation for the potential exploitation of the regulation of SIRT3 and the p53/TIGAR axis, especially targeting a reduction in p53/TIGAR to ameliorate or reverse diabetic cardiomyopathy.

### 5.1. Novelty and Significance

#### 5.1.1. What Is New?

High glucose upregulated PHDs, thus resulting in an impairment of hypoxic signaling and an increased oxygen consumption rate (OCR) in H9c2 cells, which may contribute to diabetic cardiomyopathy.High glucose upregulated p53 and TIGAR expression in cardiomyocytes. Pretreatment with p53 and TIGAR siRNA reversed the impairment of hypoxic signaling and decreased oxygen consumption rate (OCR) in cardiomyocytes.SIRT3 suppressed p53 and blunted diabetes- and hyperglycemia-induced disruption of PHDs-HIF-α signaling.

#### 5.1.2. What Is Relevant?

This is the first time it has been demonstrated that hyperglycemia modulates diabetic PHDs-HIF-α and metabolic reprogramming by a mechanism involving the upregulation of p53 and TIGAR. Targeting p53/TIGAR axis-mediated hypoxic signaling may be a novel therapeutic target for diabetic cardiomyopathy.

## Figures and Tables

**Figure 1 cells-12-01060-f001:**
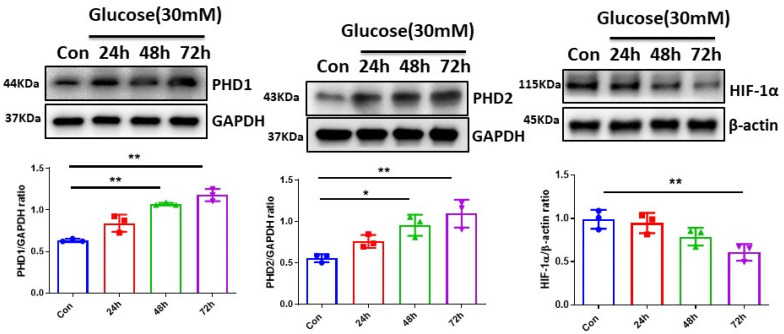
Time course of high-glucose (HG) upregulated PHDs expression and reduced HIF-1α expression in H9c2 cells. Western blot analysis demonstrating that exposure of H9c2 cells to HG (30 mM) for 24, 48, and 72 h resulted in a gradual increase in expression of PHD1 and PHD2 as compared to control normal glucose (5.5 mM). Densitometries showed a significant upregulation of PHD1 and PHD2 expression at 48 h and 72 h. Exposure of H9c2 cells to HG (30 mM) for 24, 48, and 72 h resulted in a gradual reduction in HIF-1α expression as compared to control normal glucose (5.5 mM). Densitometries showed a significant downregulation of HIF-1α expression at 72 h. All data represent mean ± SD (n = 3 per group, * *p* < 0.05 and ** *p* < 0.01). Circle, square and triangle were denoted the different experiment groups as indicated in the figures.

**Figure 2 cells-12-01060-f002:**
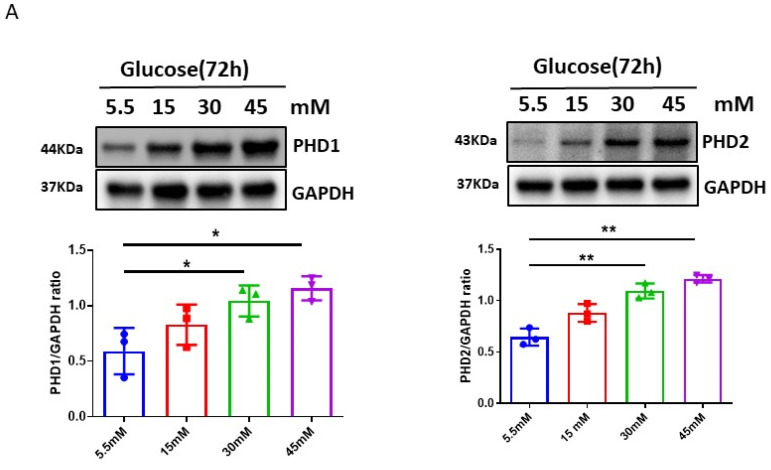
Dose response of high-glucose (HG)-induced upregulation of PHDs expression and reduction in HIF-α expression in H9c2 cells. (**A**) Western blot analysis revealing that exposure of H9c2 cells to different concentrations of glucose (15, 30, 45 mM) for 72 h resulted in a dose-dependent increase in expression of PHD1 and PHD2 as compared to normal glucose (5.5 mM). Densitometries showed a significant upregulation of PHD1 and PHD2 expression at 30 mM and 45 mM. (**B**) Exposure of H9c2 cells to different concentrations of glucose (15, 30, 45 mM) for 72 h resulted in a dose-dependent reduction in HIF-1α and HIF-2α expression as compared to normal glucose (5.5 mM). Densitometries showed a significant reduction of HIF-1α and HIF-2α expression at 30 mM and 45 mM. (**C**) Exposure of H9c2 cells to different concentrations of glucose (15, 30, 45 mM) for 72 h resulted in a gradual increase in expression of TGF-β, but a significant reduction in Ang-1 expression as compared to normal glucose (5.5 mM). Densitometries showed that the expression of Ang-1 and TGF-β1 was significant differences at 30 mM and 45 mM concentrations. All data represent mean ± SD (n = 3 per group, * *p* < 0.05 and ** *p* < 0.01). Circle, square and triangle were denoted the different experiment groups as indicated in the figures.

**Figure 3 cells-12-01060-f003:**
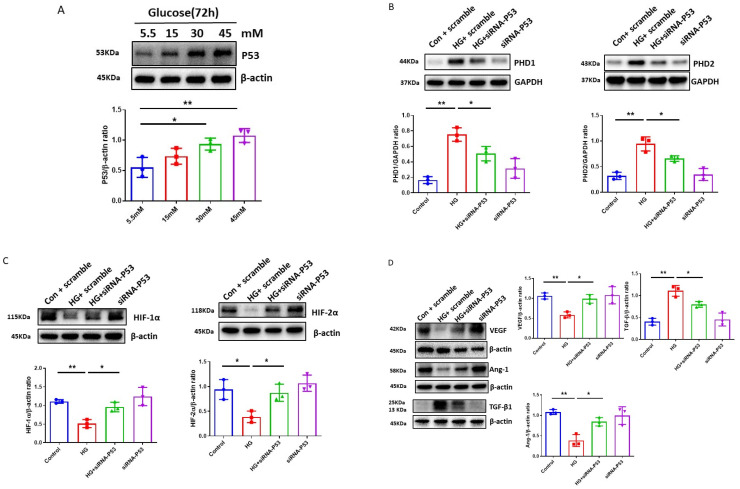
Knockdown of p53 ameliorates HG-induced disruption of PHDs-HIF-α signaling and improves metabolic metabolism in H9c2 cells. (**A**) Exposure of H9c2 cells to different concentrations of glucose (15, 30, 45 mM) for 72 h resulted in a dose-dependent increase in p53 levels as compared to normal glucose (5.5 mM). Densitometries showed a significant upregulation of p53 expression at 30 mM and 45 mM concentrations. Data represent mean ± SD (n = 3 per group, * *p* < 0.05 and ** *p* < 0.01). (**B**) Knockdown of p53 resulted in suppression of HG (30 mM)-induced upregulation of PHD1 and PHD2. Data represent mean ± SD (n = 3 per group, * *p* < 0.05 and ** *p* < 0.01). (**C**) Knockdown of p53 increased the expression of HIF-1α and HIF-2α in H9c2 cells under HG (30 mM) conditions. Data represent mean ± SD (n = 3 per group, * *p* < 0.05 and ** *p* < 0.01). (**D**) Knockdown of p53 upregulated the expression of Ang-1 and VEGF and downregulated TGF-β expression under HG (30 mM) conditions. Data represent mean ± SD (n = 3 per group, * *p* < 0.05 and ** *p* < 0.01). (**E**) Knockdown of p53 attenuated HG (30 mM)-induced increases in basal respiration, maximal respiration, spare respiration, non-mitochondrial respiration, and proton leak in H9c2 cell lines. All data represent mean ± SD (n = 5, * *p* < 0.05 and ** *p* < 0.01). (**F**) Knockdown of p53 rescued HG (30 mM)-induced impairments of glycolytic capacity and glycolytic reserve in H9c2 cell lines. All data represent mean ± SD (n = 5, ** *p* < 0.01). Circle, square and triangle were denoted the different experiment groups as indicated in the figures.

**Figure 4 cells-12-01060-f004:**
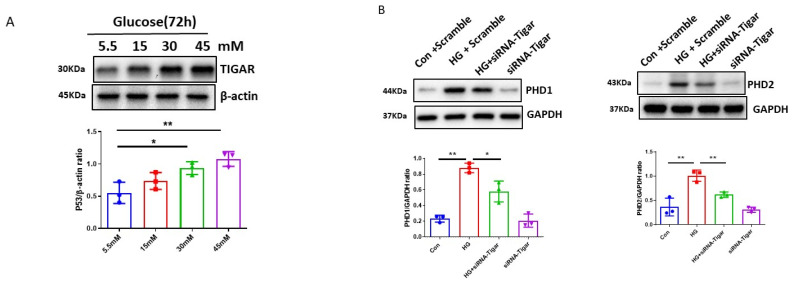
Knockdown of TIGAR improves cardiomyocyte PHDs-HIF-α signaling under HG conditions. (**A**) Western blot analysis revealing that exposure of H9c2 cells to different concentrations of glucose (15, 30, 45 mM) for 72 h resulted in a dose-dependent increase in TIGAR expression as compared to normal glucose (5.5 mM). Densitometries showed a significant upregulation of TIGAR expression at 30 mM and 45 mM concentrations. Data represent mean ± SD (n = 3 per group, * *p* < 0.05 and ** *p* < 0.01). (**B**) Knockdown of TIGAR blunted HG (30 mM)-induced upregulation of PHD1 and PHD2. Data represent mean ± SD (n = 3 per group, * *p* < 0.05 and ** *p* < 0.01). (**C**) Knockdown of TIGAR reversed HG (30 mM)-induced downregulation of HIF-1α and HIF-2α expression. Data represent mean ± SD (n = 3 per group, ** *p* < 0.01). (**D**) Immunostaining revealing that knockdown of TIGAR significantly reduced FSP-1 levels in H9c2 cells under HG (30 mM) conditions (20X Magnification). All data represent mean ± SD (n = 3 per group, ** *p* < 0.01). (**E**) Knockdown of TIGAR attenuated HG (30 mM)-induced increases in basal respiration, maximal respiration, spare respiration, non-mitochondrial respiration, and proton leak in H9c2 cell lines. All data represent mean ± SD (n = 5, ** *p* < 0.01). Circle, square and triangle were denoted the different experiment groups as indicated in the figures.

**Figure 5 cells-12-01060-f005:**
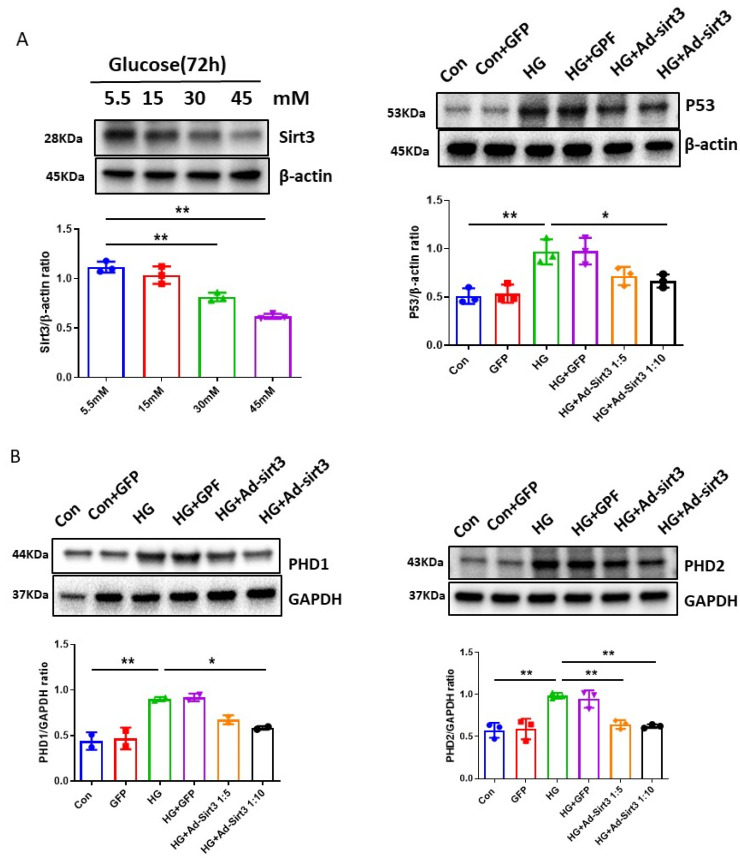
SIRT3 blunts HG-induced p53 expression and modulates hypoxic signaling in H9c2 cells. (**A**) Exposure of H9c2 cells to different concentrations of glucose (15, 30, 45 mM) for 72 h resulted in a dose-dependent reduction in SIRT3 expression as compared to normal glucose (5.5 mM). Densitometries showed a significant reduction of SIRT3 expression at 30 mM and 45 mM concentrations. Ad-SIRT3 treatment significantly attenuated the HG (30 mM)-induced p53 expression. All data represent mean ± SD (n = 3 per group, * *p* < 0.05 and ** *p* < 0.01). (**B**) Ad-SIRT3 treatment blunted HG (30 mM)-induced upregulation of PHD1 and PHD2. Data represent mean ± SD (n = 3 per group, * *p* < 0.05 and ** *p* < 0.01). (**C**) Ad-SIRT3 treatment resulted in a dose-dependent upregulation of HIF-1α expression in H9c2 cells under HG (30 mM) conditions. Data represent mean ± SD (n = 3 per group, ** *p* < 0.01). (**D**) Levels of FSP-1 protein immunostaining revealing that Ad-SIRT3 treatment significantly reduced FSP-1 levels in cultured H9c2 cells under HG (30 mM) conditions (20X Magnification). All data represent mean ± SD (n = 3 per group, ** *p* < 0.01). Circle, square and triangle were denoted the different experiment groups as indicated in the figures.

**Figure 6 cells-12-01060-f006:**
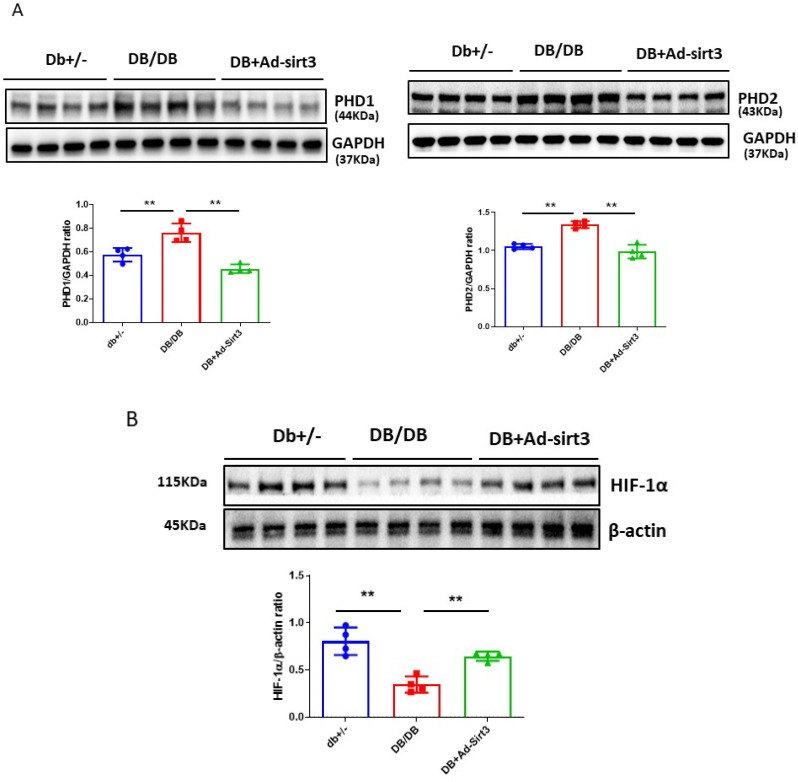
SIRT3 regulates hypoxic signaling in the hearts of diabetic db/db mice. (**A**) Western blot and densitometric analysis demonstrating that expression of PHD1 and PH2 were significantly increased, while treatment with Ad-SIRT3 (1 × 10^9^ PFU) significantly reduced the expression of PHD1 and PHD2 in the hearts of diabetic db/db mice. (**B**) Western blot and densitometric analysis showing that the expression of HIF-1α was significantly reduced, while treatment with Ad-SIRT3 (1 × 10^9^ PFU) significantly upregulated the expression of HIF-1α in the hearts of diabetic db/db mice. (**C**) Representative images and quantitative analysis showing that treatment with Ad-SIRT3 significantly reduced myocardial FSP-1 levels in db/db mice by FSP-1 immunostaining (Red, 10X Magnification). All data represent mean ± SD (n = 4 per group, ** *p* < 0.01). Circle, square and triangle were denoted the different experiment groups as indicated in the figures. (**D**). Novel role of p53/TIGAR signaling pathway in the regulation of hyperglycemia and diabetes-induced metabolic programming by a mechanism involvement of disruption of PHDs/HIFs. High glucose activates prolyl hydroxylases and disrupts HIF-α signaling, which alters cell metabolism by a reduction in glycolysis while increasing the oxygen consumption rate in cardiomyocytes. Mechanistically, knockdown of p53/TIGAR increased glycolysis under high glucose conditions and suppressed HG-induced increases in oxygen consumption rate. Therefore, targeting the p53/TIGAR axis may alleviate hyperglycemia-induced cardiomyopathy via a mechanism involving the hypoxic signaling pathway and reprogrammed cell metabolism.

## Data Availability

The authors declare that all supporting data are available within the article.

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
