# Peer review of "High Glucose Activates Prolyl Hydroxylases and Disrupts HIF-α Signaling via the P53/TIGAR Pathway in Cardiomyocyte"

_cells, 2023, doi:10.3390/cells12071060_

Round 1

Reviewer 1 Report

The authors presented a novel and interesting information about the effect of high glucose on PHDs-HIF-a signaling pathway. They reported that high glucose disrupted PHDs-HIF-a signaling pathway and altered cell metabolism by a mechanism involving the p53 and TIGAR upregulation and the SIRT3 downmodulation. They conducted an in vitro study on the H9C2 using loss and gain of functions experiments and the in vivo study on diabetic mice. Although the paper is nicely written and reports interesting findings, it cannot be accepted for publication in MDPI IJMS in its present form.

Major comments:

1) The authors should better explain the combined treatment of HG and siRNAs (for p53 and TIGAR). How long does siRNA treatment last? When is glucose added? During, after or before the siRNA treatment?

2) Figure 3 and 4 do not show the protein level of the silenced genes (p53 and TIGAR) at the indicated treatments, especially for “HG plus scrambled” and “HG plus siRNA” which are also not present in the previous publication (reference 37). The authors should demonstrate the effective reduction in p53 and TIGAR expression by the combine (siRNA plus HG) compared to the single treatment (HG plus scrambled). Eventually, the authors should also analyze the transcriptional expression of the silenced genes.

3) The same problem occurs for the experiment performed with Ad-sirt3. Is Ad-sirt3 infection antecedent, concomitant or following the high glucose treatment? Is Sirt3 protein more enriched in the HG+Ad-Sirt3 sample compared to HG+GFP? For the reason explained in point 2 and 3, this reviewer is of the opinion that the in vitro results do not fully support the conclusions drawn by the authors that “knockdown of p53/TIGAR or overexpression of SIRT3 in cardiomyocytes blunted HG-induced impairment of HIF-α signaling, and reprogrammed cellular metabolism by reducing the oxygen consumption rate”.

4) The authors conducted the Sirt3 transgenic expression in male diabetic mice. It is not clear why the authors chose this sex, not both males and females. The authors should indicate in the material and methods how many mice have been treated. Finally, Adenovirus needs to be explained in more detail. Does it contain a tissue-specific promoter?

Minor comments:

1) The molecular weight of the proteins should be added in the figures showing the western blots.

2) Please, replace “transfected” (lane 118 and 311) and “transfection” (lane 312) with “ infected” and “infection”.

3) Is not clear the adenovirus‐β‐gal vector used in the in vivo study (see material method lane 86).

Author Response

Reviewer 1:

1) The authors should better explain the combined treatment of HG and siRNAs (for p53 and TIGAR). How long does siRNA treatment last? When is glucose added? During, after or before the siRNA treatment?

Thank the review for this comment. We knocked down p53 and TIGAR first, then these KO cells were incubated with HG (30mM) for 72 hours. We included this information in revision methods.

2) Figure 3 and 4 do not show the protein level of the silenced genes (p53 and TIGAR) at the indicated treatments, especially for “HG plus scrambled” and “HG plus siRNA” which are also not present in the previous publication (reference 37). The authors should demonstrate the effective reduction in p53 and TIGAR expression by the combine (siRNA plus HG) compared to the single treatment (HG plus scrambled). Eventually, the authors should also analyze the transcriptional expression of the silenced genes.

We thank the reviewer for these comments. We knocked down p53 and TIGAR first, then exposed KO cells to high glucose conditions. We did not check these proteins after exposure to HG since p53 or TIGAR are knocked down in these cells. We only measured protein expression and did not check the transcriptional mRNA levels. In our revision, we have changed our description regarding knockdown or overexpression, replacing with “pretreatment of p53/TIGAR siRNA or Ad-SIRT3 pretreatment in cardiomyocytes”

3) The same problem occurs for the experiment performed with Ad-sirt3. Is Ad-sirt3 infection antecedent, concomitant or following the high glucose treatment? Is Sirt3 protein more enriched in the HG+Ad-Sirt3 sample compared to HG+GFP? For the reason explained in point 2 and 3, this reviewer is of the opinion that the in vitro results do not fully support the conclusions drawn by the authors that “knockdown of p53/TIGAR or overexpression of SIRT3 in cardiomyocytes blunted HG-induced impairment of HIF-α signaling, and reprogrammed cellular metabolism by reducing the oxygen consumption rate”.

Similarly, we infected cells with adenovirus SIRT3 first, then exposed of these cells to high glucose. We did not check these proteins after exposure to HG since SIRT3 is increased in these cells. Although we did not check levels of these genes under HG conditions, our data do support our overall conclusion that pretreatment of p53/TIGAR siRNA or Ad-SIRT3 pretreatment in cardiomyocytes blunted HG-induced impairment of HIF-α signalling, and reprogrammed cellular metabolism by reducing the oxygen consumption rate. Please see our revision.

4) The authors conducted the Sirt3 transgenic expression in male diabetic mice. It is not clear why the authors chose this sex, not both males and females. The authors should indicate in the material and methods how many mice have been treated. Finally, Adenovirus needs to be explained in more detail. Does it contain a tissue-specific promoter?

We chose male diabetic mice because they have significant cardiac dysfunction at this age as compared to females which have only mild cardiac dysfunction. The adenovirus vector is not tissue specific, we have discussed this in the previous study (ref 37). Ad-SIRT3 was mainly expressed in cardiomyocyte and endothelial cells of the diabetic db/db mouse hearts. We cited the previous study regarding the expression of SIRT3 on which cells in the heart in the revised methods.

Minor comments:

1) The molecular weight of the proteins should be added in the figures showing the western blots.

Thank the review for this comment. We added the molecular weight of each proteins that were tested in WB. Please see our revised Figures.

2) Please, replace “transfected” (lane 118 and 311) and “transfection” (lane 312) with “ infected” and “infection”.

These were corrected following the reviewer’s suggestion.

3) Is not clear the adenovirus‐β‐gal vector used in the in vivo study (see material method lane 86).

In this study, only d/db mice did Ad-SIRT3 treatment, we did not use adenovirus control vector.

Reviewer 2 Report

The original research manuscript titled “High glucose activates prolyl hydroxylases and disrupts Hif-α signaling via the p53/tigar pathway in cardiomyocyte” demonstrates that hyperglycemia modulates PHDs-HIF-α and metabolic reprogramming by a mechanism involving upregulation of p53 and TIGAR. Thus, targeting p53/TIGAR axis-mediated hypoxic signaling may be a novel therapeutic target for diabetic cardiomyopathy.. The manuscript digs deep into the various strategies reported across scientific literature and presents a gripping story towards understanding of a novel signalling process.

Major points:

1.     A schematic flowchart of the signalling process might be helpful to understand the process of the regulation in the hyperglycemia-induced disruption of PHDs-HIF-α signalling.

2. The main question addressed by the research is that hyperglycemia and diabetes modulate hypoxic signaling to disrupt HIF1a-PHD via downregulation of SIRT3 and upregulation of p53/TIGAR. The only gap it addresses is the role of PHD1 during hyperglycaemia and in oxygen consumption rate. In that respect, a definite correlation of PHD1 with metabolic data like OCR was important. That is lacking in the current manuscript.

The db/db mice used in the study were treated with Sirt3-adenovirus. Another experimental group with knockdown of PHD1 or PHD2 might be interesting towards understanding of the process. The mice body weight, heart weight and glucose levels along with cardiac functional data might be of interest in the understanding of the study.

3. The role of the sensor in PHD1 during diabetic cardiomyopathy may be of interest towards regulation of cell metabolism.

4. A definite correlation of PHD1 with metabolic data like OCR was important. That is lacking in the current manuscript. The db/db mice used in the study were treated with Sirt3-adenovirus. Another experimental group with knockdown of PHD1 or PHD2 might be interesting towards understanding of the process. The mice body weight, heart weight and glucose levels along with cardiac functional data might be of interest in the understanding of the study.

The siRNA used in the study have not been compared with scramble siRNA.

5. The conclusions might be further supported with the given comments

6. The signalling seems to be represented in the form of a graphical abstract. So many molecules have been discussed in the manuscript. A more focussed approach on PHD1 might have been more interesting.

Author Response

Reviewer 2

Major points:

  1. A schematic flowchart of the signalling process might be helpful to understand the process of the regulation in the hyperglycemia-induced disruption of PHDs-HIF-α signalling.

Thank the reviewer for this constructive comment. This is done following reviewer’s suggestion. Please see new Figure 6D.

  1. The main question addressed by the research is that hyperglycemia and diabetes modulate hypoxic signaling to disrupt HIF1a-PHD via downregulation of SIRT3 and upregulation of p53/TIGAR. The only gap it addresses is the role of PHD1 during hyperglycaemia and in oxygen consumption rate. In that respect, a definite correlation of PHD1 with metabolic data like OCR was important. That is lacking in the current manuscript.

The db/db mice used in the study were treated with Sirt3-adenovirus. Another experimental group with knockdown of PHD1 or PHD2 might be interesting towards understanding of the process. The mice body weight, heart weight and glucose levels along with cardiac functional data might be of interest in the understanding of the study.

We agree with reviewer’s comments. It is important to define the role of PHD1 or PHD2 in metabolism of cardiomyocytes. Our present study was mainly focused on the role of p53/TIGAR in PHD1,2/HIFs pathways. However, the potential roles of PHD1&PHD2 in cardiomyocyte metabolism are beyond the scope of our present study and warrant further investigation.

  1. The role of the sensor in PHD1 during diabetic cardiomyopathy may be of interest towards regulation of cell metabolism.

We agree with reviewer’s comments, but these studies are beyond the scope of our present study and warrant further investigation.

  1. A definite correlation of PHD1 with metabolic data like OCR was important. That is lacking in the current manuscript. The db/db mice used in the study were treated with Sirt3-adenovirus. Another experimental group with knockdown of PHD1 or PHD2 might be interesting towards understanding of the process. The mice body weight, heart weight and glucose levels along with cardiac functional data might be of interest in the understanding of the study.

The impact of Ad-SIRT3 on body weight, heart weight, and glucose levels were published in our previous study (ref 37). In the present study, we did not examine the roles of PHD1 and PHD2 in the diabetic heart. These studies warrant further investigation. We added discussion in our revision.

The siRNA used in the study have not been compared with scramble siRNA.

 We included scramble siRNA.

  1. The conclusions might be further supported with the given comments

We thank the reviewer for these constructive comments. We took these comments to the heart and have included and discussed these comments in our revised paper.

  1. The signalling seems to be represented in the form of a graphical abstract. So many molecules have been discussed in the manuscript. A more focussed approach on PHD1 might have been more interesting.

We thank the reviewer for these comments and have added a graphical abstract. Please see new Figure 6D.

Round 2

Reviewer 1 Report

I thank authors for clearly addressing all my points. The manuscript has gained improvements and quality, following the revision and response to the points. Therefore, the manuscript is now suitable to be accepted in the current stage.